# Blind cavefish evolved higher foraging responses to chemo- and mechanostimuli

**Kyleigh Kuball[1], Vânia Filipa Lima Fernandes[2], Daisuke Takagi[3], Masato Yoshizawa[1] \***

**1** School of Life Sciences, The University of Hawai'i at Mānoa, Honolulu, Hawai'i, United States of America, **2** Institut de Biologie Valrose, Université Côte d'Azur, Nice, France, **3** Department of Mathematics, The University of Hawai'i at Mānoa, Honolulu, Hawai'i, United States of America

\* yoshizaw@hawaii.edu

**Data Availability Statement:** Excel macro script is available at https://zenodo.org/record/7996590. The video datasets generated and/or analyzed during the current study are available at the university's shared server and will be deposited to Zenodo (https://zenodo.org/).

## Abstract

In nature, animals must navigate to forage according to their sensory inputs. Different species use different sensory modalities to locate food efficiently. For teleosts, food emits visual, mechanical, chemical, and/or possibly weak-electrical signals, which can be detected by optic, auditory/lateral line, and olfactory/taste buds sensory systems. However, how fish respond to and use different sensory inputs when locating food, as well as the evolution of these sensory modalities, remain unclear. We examined the Mexican tetra, *Astyanax mexicanus*, which is composed of two different morphs: a sighted riverine (surface fish) and a blind cave morph (cavefish). Compared with surface fish, cavefish have enhanced non-visual sensory systems, including the mechanosensory lateral line system, chemical sensors comprising the olfactory system and taste buds, and the auditory system to help navigate toward food sources. We tested how visual, chemical, and mechanical stimuli evoke food-seeking behavior. In contrast to our expectations, both surface fish and cavefish did not follow a gradient of chemical stimulus (food extract) but used it as a cue for the ambient existence of food. Surface fish followed visual cues (red plastic beads and food pellets), but, in the dark, were likely to rely on mechanosensors—the lateral line and/or tactile sensor —as cavefish did. Our results indicate cavefish used a similar sensory modality to surface fish in the dark, while affinity levels to stimuli were higher in cavefish. In addition, cavefish evolved an extended circling strategy to forage, which may yield a higher chance to capture food by swimming-by the food multiple times instead of once through zigzag motion. In summary, we propose that ancestors of cavefish, similar to the modern surface fish, evolved extended food-seeking behaviors, including circling motion, to adapt to the dark.

## Introduction

Many teleost species rely on visual information for foraging, although fishes employ a wide range of sensory modalities for foraging strategies [1–4]. These strategies range from drift-hunting by coelacanths that use a single sensory modality (electroreception) to detect benthic prey [5], to the multi-sensory, active pursuit of prey by bonnethead sharks, which use long-distance olfactory signals followed by visual cues to precisely locate prey [2].

**Funding:** MY: National Institute of Health/ National Institute of General Medical Sciences (GM125508 and GM148960) MY: Hawaii Community Foundation (18CON-90818) DT: United States Army Research Office (W911NF-17-1-0442) The funders had no role in study design, data collection and analysis, decision to publish, or preparation of the manuscript.

**Competing interests:** The authors have declared that no competing interests exist.

Given the breadth of sensory systems, how the coordination and hierarchical use of sensory systems change during the adaptation to a new environment remains unclear. Depending on species, different mechanisms are favored, such as mechano-, chemo-, and/or electro-sensing [1, 2]. For foraging tradeoffs between finding (energy loss) and consuming food gains (energy gain), animals should strategize to maximize energy gain with minimum loss by leveraging available sensory inputs [6]. To tackle this question, we chose the freshwater Mexican tetra, *Astyanax mexicanus*. *Astyanax mexicanus* is a ~6 cm freshwater fish, consisting of two morphs: riverine and sighted surface form (surface fish: colonizing in a rage of south Texas USA to the south American continent) and the cave-dwelling blind form (cavefish: limestone mountain ranges at Northeast Mexico). We then conducted foraging experiments comparing these different populations of the same species.

Cavefish-dwelling cave environment is typically food sparse and perpetually dark. Food availability is thought to be changed during the cycle of rainy and dry seasons (approx. 6 months and 6 months, respectively, per year), and limited food input is thought during the dry season [7]. In contrast, surface fish live in food-abundant environments whole year. Both cave and surface fish are omnivores [8]. Upon this ecological condition, cavefish show higher responses to mechanical vibration stimulus at ~40 Hz than surface fish. The 40 Hz vibration can be typically generated by crawling crustaceans [9], and cavefish enhanced this response by the increased cranial mechanosensory lateral line [10, 11]. Fish with higher vibration responses, called vibration attraction behavior (VAB) dominated over prey capture in the dark [12, 13]. Cavefish also have finer chemical sensing, such as the ability to respond to $10^5$ lower concentrations of amino acids than surface fish (*i.e.*, cavefish can respond to $10^{-10}$ M of alanine, whereas surface fish respond to $10^{-5}$ M of it or higher) [14]. In contrast, no detectable difference in auditory response has been reported between surface fish and cavefish [15] and there is no comparative study in tactile sensing between these two morphs (but see Voneida & Fish [16]).

Upon this powerful comparative model system, it remains largely unknown how these sensory systems were strategically utilized during foraging: are these sensory systems used equally for foraging, or is there any hierarchical order of the usage of the sensory systems? Then, if there is a hierarchical order, what is its ecological relevance? To provide answers to these questions, we designed experiments using varying stimuli. We used (1) water droplets as the source of mechanical stimulus (auditory only, when it hits the water surface), (2) food extract suspended in water as the source of the mechanical (auditory) + chemical stimuli—only chemical stimulus is the additional to (1), (3) red plastic beads as visual + mechanical (auditory + lateral line/tactile) stimuli, which are additional to (1), (4) food extract and plastic beads simultaneously, and (5) fish commercial diet as a positive control. We then measured latency as the initial response to these stimuli, number of foraging attempts as the proxy for robustness of foraging mode, and zigzag and circling measurements (duration and bout numbers) to characterize two foraging strategies in surface fish and cavefish. Foraging with circling is typical in cavefish; however, it was not clear if surface fish showed zigzag or circling in the dark previously (see Result and Discussion section about the behavioral characteristics of zigzag and circling).

Our result indicated that, for latency measurements, some surface fish and cavefish responded to sole auditory stimulus (water droplet) in either light or dark conditions, but their response became more robust with visual (beads: for surface fish) or chemical (food scent: for cavefish) stimulus, suggesting both fish rely on multiple sensory inputs for the initial response (latency). For example, the beads stimulus, which stimulated auditory, lateral line and tactile sensing, evoked shorter latency in cavefish. In contrast, chemical stimuli (food extract) evoked a prominent searching behavior in cavefish than the beads stimuli. In the dark, both morphs

directly aimed at the water surface or the bottom of the tank where the food usually ended up but did not aim to the highest concentration of the scent, suggesting chemical stimuli did not navigate them toward food sources but instead evoked fish to the existence of food. Cavefish showed higher foraging activities than surface fish under chemical stimulus.

In summary, surface fish tended to require multiple sensory stimuli to engage to forage. In contrast, the sole auditory/chemical stimuli were still able to induce food-searching behavior in cavefish. Among the given stimuli, chemical stimulus strongly drove food-searching behavior at the bottom of the tank and at the water surface in both surface fish and cavefish whilst the food extract plume was still at the middle of the water column, suggesting fish did not directly use chemical gradients but instead used the chemical stimulus as ambient cues of the food existance. Further, we also detected different foraging patterns between the light and dark conditions even in blind cavefish, and the differences between surface and cavefish in two diet-locating strategies—zigzag and circling. Our result provides new insight into the evolution of foraging strategies for diet-related stimuli.

## Materials and methods

### Fish maintenance and care

Populations of *A. mexicanus* (both sighted and the blind morphs) were raised and bred at the University of Hawaiʻi at Mānoa aquatic facility with care and protocols approved under IACUC (17–2560) at University of Hawaiʻi at Mānoa. Both surface fish and cavefish were *Astyanax mexicanus* species. Surface fish raised in the lab were descendants from those collected by Dr. William R. Jeffery from Balmorhea Springs State Park in Texas and cavefish were descendants collected by Richard Borowsky and Dr. William R. Jeffery in Cueva de El Pachón in Tamaulipas, Mexico. Both surface fish and cavefish were raised on a 12:12 light cycle in 42-liter tanks in a custom water-flow tank system. Temperatures were maintained at 21˚C ± 0.5˚C for rearing, 24˚C ± 0.5˚C for behavior experiments, and 25˚C ± 0.5˚C for breeding. Their diet consisted of TetraColor tropical fish food granules and TetraMin tropical fish food crisps (Tetra, Blacksburg, VA) and jumbo mysis shrimp (Hikari Sales, USA, Inc., Hayward, CA). Fish were fed on Zeitgeber time 3 and 9 and maintained at 7.0 pH with a water conductivity of 600–800 µS.

### Experimental populations

We used a 37.9 L tank to house each experimental population (surface and cavefish) prior to introducing the stimuli. Three fish (N = 3) of each population were acclimated in this 37.9 L tank at least for a week, and at least four days prior to recording, fish tanks were cleaned and the tank water was replaced with conditioned fish water (pH 6.8–7.0, conductivity: ~700 µS adjusted with Reef Crystals Reef Salt, Instant Ocean, Blacksburg, VA). Fish in the cleaned tank were then placed on the recording stages in a light-controlled room where fish circadian rhythm was entrained by a 12:12 h light-dark cycle with 30–100 lux light. On recording days, the experiment commenced at ~2 hours of Zeitgeber time, which is a similar time for everyday feeding to provide a higher chance of exhibiting foraging behaviors. Fish were fed normally until a day before recording, and were not fed before and during the recording at the recording day. We set recording cameras (see below), set blackboards on the side of the arena to prevent extra visual stimulus from the side, and started recording the video for a 10-min acclimation time prior to introducing stimulus. This acclimation time video is to check any odd motions among fish before providing actual stimuli. The stimuli were administered in the following order: (1) water droplets (3 drops), (2) red plastic beads (4.7 mm diameter: Millipore Sigma, Burlington, MA), (3) food extract (see below), (4) a combination of food extract & beads, and

(5) agar-solidified food (see below). Each of the stimuli were given in 10-min intervals in this order. Recording was performed for ~60 min in total. The dark experiment (no light) and the light experiment (30–100 lux) were performed on different days. We repeated the above experiment twice (N = 3 fish × 2 experiments = 6 in total per population) and used 12 adult fish (2 populations) in total.

## Experimental stimulus

The water stimulus was three droplets of distilled water with 0.1% Methylene Blue (Millipore-Sigma). The beads stimulus was 4–5 of red polystyrene beads (4.7mm in diameter). The food extract was made by suspending 0.1 g of fine ground Tropical XL Color Granules with Natural Color Enhancer (Tetra U.S., Blacksburg, VA) in 2 mL of distilled water mixed with 0.5 mL of 0.5% Methylene Blue (MilliporeSigma) and filtered with a 0.45 μm syringe filter. The food extract was made fresh for each experiment and three drops were added as the stimulus. The agar-solidified food was comprised of 1.0 g of fine ground Tropical XL Color Granules with Natural Color Enhancer (red colored granules) suspended with 5 mL of 1% agar (Millipore-Sigma) in the fish conditioned water (pH 6.8–7.0, conductivity ~700 μS), then poured into 6-cm dishes to solidify. Once solidified, a razor blade sterilized with 70% ethanol was used to cut the agar food into 5 × 5 mm squares and 3–4 pieces were given per stimulus. Sinking of red plastic beads was approximately the same as the red agar food, mimicking red agar food movement.

## Recordings

All light condition videos were recorded on an iPhone Xs (Apple, Cupertino, CA) at 30 fps. Fish behaviors in the dark were recorded using a custom-made infrared back-light system (SMD 3528 850nm strip: LightingWill, Guang Dong, China). A LifeCam studio 1080p HD webcam (Microsoft, Redmond, WA, USA) with a zoom lens (Zoom 7000, Navitar, Rochester, NY, USA) fitted with an IR high-pass filter (Optical cast plastic IR long-pass filter, Edmund Optics Worldwide, Barrington, NJ, USA). This USB webcam (LifeCam studio 1080p HD webcam) was used to record at 20 fps using virtual dub software (version 1.10.4, http://www.virtualdub.org/). Once recorded, videos were uploaded to Google Drive for accessibility.

## Video analysis

Videos were analyzed using Behavioral Observation Research Interactive Software (BORIS V. 7.4.11-2019-02-28, Department of Life Sciences & Systems Biology, University of Torino-Italy). For video analysis, the tank was divided into nine square sections, with areas 1, 2, 3, and 5 as the top row and areas 7–9 as the bottom (Fig 1B, the far-left panel). Using BORIS, each fish's actions were recorded during the videos. Latency was defined as the measurement of time duration between when stimulus hit the water surface and when fish of interest approached at the dropping point. "Attempts" were measured as the number of capturing or biting motion against the stimulus by observing the opening and closing of the mouth rapidly or picking up a bead/food. A "zigzag" motion was defined as rapid changes of the swimming direction every ~ 1 s or less, and was measured as occurrence (bout number) and duration (s). "Circling" motion was defined as the continuous unidirectional turnings without glide swimming, and was measured as occurrence (bout number) and duration (s) by unidirectional turning to make at least one full circle at the tank bottom or water surface.

We recorded the tank areas where each behavior was observed. Quantitative data collected from BORIS was then consolidated in the Excel macro (Microsoft, Redmond, WA) (https://zenodo.org/record/7996590).

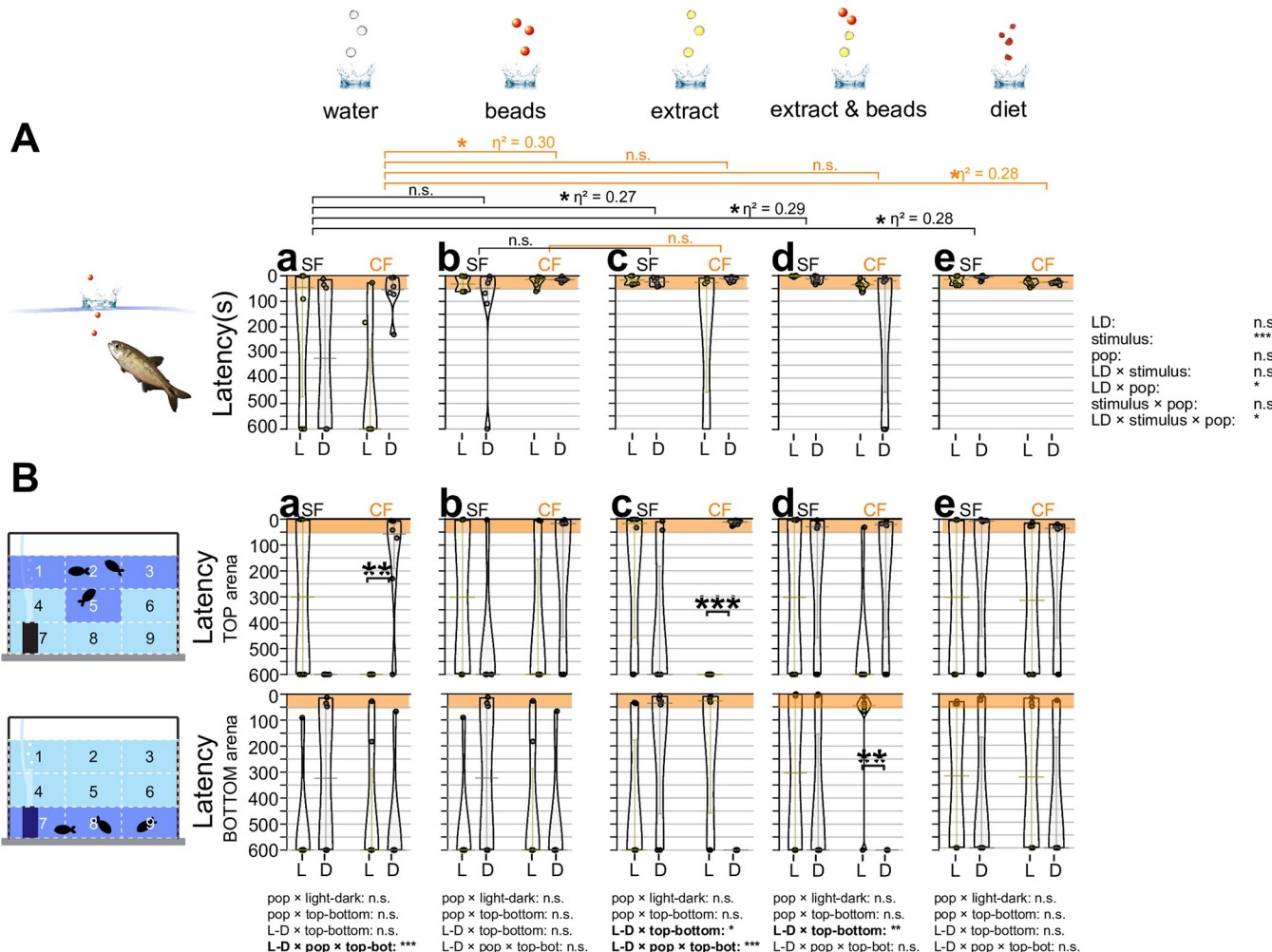

**Fig 1. Latencies in response times to different sensory stimuli.** (**A**) Overall latency (s) between when the object hit the water surface and when fish directly aimed toward the object. Three fish in a tank were given three droplets of reverse-osmosis (RO) purified water (water: panels **Aa** and **Ba**), three red plastic beads 4.7 mm in diameter (beads: **Ab** and **Bb**), three droplets of food extract (extract: **Ac** and **Bc**), three droplets of food extract followed by three red beads (extract and beads: **Ad** and **Bd**), and 3–4 granules (3–5 mm in diameter) of actual food (diet: **Ae** and **Be**; see Materials and Methods). (**A**) Latencies of surface fish (SF: left) and cavefish (CF: right) are shown on the y-axis. Top: shorter latency; bottom: no response within a 10 min observation (600 s). Latencies under light conditions (L: yellow bars and dots) and dark conditions (D: gray bars and dots) are also shown. The first 60 seconds after the object hit the water surface are shaded red. The statistical test results of the generalized linear model are shown on the far right. For each comparison, light and dark conditions were compared within the population per treatment (e.g., a bracket in CF with the water stimulus). Within each population, different stimuli were compared with the water stimulus and significances were calculated via Mann-Whitney tests adjusted by Holm's correction, shown as brackets at the top of boxes. All comparisons were non-significant (n.s.) in latencies. (**B**) Fish locations were tracked as the top (top row) or bottom (bottom row) and measured latencies as the same as Fig 1A. The far-left panels indicate the areas counted as the top (areas 1, 2, 3 and 5), and the bottom (areas 7, 8 and 9). The y-axes and brackets in **Ba**-**Be** represent the same as (**A**). All stars represent P-values after Holm's correction. Statistical test summaries using the generalized linear model including arena locations (top-bottom) are shown at the bottom of the boxes. Only interaction results are shown. Details of all statistics scores in this figure are found in S1 Table. n.s.: not significant, *: P < 0.05, **: P < 0.01, ***: P < 0.001.

## Statistical analysis

Quantitative data were exported from BORIS to Excel. Using macros in Excel, data were compiled and the totals of each foraging behavior were calculated (shared on Zenodo: https://zenodo.org/record/7996590). All statistical analyses were performed in RStudio 2023.12.0 (RStudio, Boston, MA, USA) with R (version 4.3.1 [17]). The R packages used included *lme4*, *lmerTest*, *car*, *coin*, *yarrr*, *ggplot2*, *AICcmodavg*, and *ggpubr*. Linear or generalized linear models (family = Gamma or Poisson) were selected using Akaike's information criterion (AIC)

function to identify the best fit models for analyses for latency, attempt, and zigzag and circling motions. We used multifactorial variance analyses using generalized linear model fitting functions (glm or glmer in the *lme4* package). Post-hoc tests were performed using the linear model (*t*-test) or the Wilcoxon signed-rank test, according to the AIC-selected model—parametric or non-parametric—, respectively. P-values were adjusted by Holm's multiple-test correction. Effect sizes were calculated through eta ($\eta^2$: eta_squared() function) or r (wilcox_-effsize() function) in the rstatix package v0.7.2 [18]

## Results and discussion

Foraging attempt was composed of initial investigation (measured by latency), adherence to the stimulus source (proxy of the number of attempts) and searching mode (zigzag or circling motion) to analyze differences in foraging strategies between surface fish and cavefish. We consider the water droplet stimulus as the baseline of the fish response when something introduced in the tank, and compared it with other stimuli.

### Latency

For the response to the water droplet stimulus, there was no detectable difference between surface fish and cavefish (water droplets; Fig 1Aa and S1 Table). Detailed scoring by discriminating the fish tank areas revealed that cavefish were attracted to water droplet stimulus when droplets hit the water surface (top) in the dark (Fig 1Ba and S1 Table). In contrast, under light conditions, cavefish responded less to the water droplet. Since cavefish seem to sense ambient light with brain opsins [19] and the lighted conditions pose an exposure risk to their predators in the wild [20], cavefish may have a reserved response under light conditions. Surface fish did not consistently respond to water droplets (Fig 1Aa and 1Ba), suggesting auditory stimulus was not sufficient to evoke robust foraging behaviors. The water droplet stimulus could be detected by the inner ear and less likely by the mechanosensory lateral line system because, typically, the lateral line could sense ~1.5× of the body length distance [21], which is ~ 10 cm away from *Astyanax* fish. The inner ear can sense a much further distance [22].

For beads, which potentially stimulate visual, auditory (when it hit the water surface), and tactile (when fish touched it at the bottom) sensors, surface fish responded quickly (~10 s) by swimming toward the top and toward the bottom of the arena under light and dark conditions, respectively (Fig 1Ab and 1Bb), although we failed to detect the significant difference between beads and water stimuli (Fig 1Aa and 1Ab). Cavefish responded to beads similarly to surface fish in the dark irrespective of light or dark conditions (Fig 1Bb), suggesting surface fish and cavefish used similar sensory modalities in initial responses against solid food-like objects in the dark.

The food extract showed somewhat similar latency response to water droplets but triggered more engagement toward the bottom (surface fish in the light and dark and cavefish in the light) or the top of the tank (cavefish in the dark) (Fig 1Ac and 1Bc). Importantly, food extract always dispersed in the middle of the recording tank and the dense food-extract plume (dyed with methylene blue; see Materials and Methods; S1 Movie) never reached the bottom before dispersing, suggesting chemical stimulus was not used to orient food location, but may be used as a signal of food existence in a given environment (ambient existence). Cavefish aiming at the top of the tank in the dark could be explained similarly to that evoked by water droplets (i.e., boldness in the dark; see above), however cavefish significantly responded and aimed to the bottom in the lighted condition, which was not observed with the water droplet stimulus (Fig 1Bc).

The combined bead and food-extract stimulus (Fig 1Ad) invoked the intermediate response of beads-only and food extract-only stimulus in cavefish, where cavefish responded to the stimulus by aiming to the bottom under the light condition and aimed at either the top or bottom under the dark condition (Fig 1Bd). Surface fish were engaged toward either the top or bottom under the light and dark conditions, which were similar to the responses to the food stimulus (Fig 1Bd and 1Be). Cavefish aimed at either the top or bottom with food stimulus and no notable difference in the forage manners was detected compared with the food extract (Fig 1Bc and 1Bd).

In summary, water droplet stimulus (auditory) evoked a light-dependent response in the blind cavefish, whereby dark conditions seemed to make cavefish more explorative to come to the water surface. Other stimuli induced different light- and area-dependent responses in surface fish and cavefish, but opposite responses: surface fish foraged in the light, but cavefish foraged in the dark, assuming attraction to the top area as a bolder response. However, overall latencies were similar between surface fish and cavefish in different stimuli and under dark conditions (Fig 1A), suggesting cavefish did not evolve particular sensory responses during initial foraging attempts (latency) in the dark.

## Number of foraging attempts

Fish attempted to bite or capture the stimulus source following initial contact. We measured this engagement to foraging defined by darting/thrusting and biting motions against the stimulus source (i.e., attempts). In contrast to the initial response (*i.e.*, latency), water droplets did not evoke any attempts in either surface fish or cavefish in either light or dark conditions (Fig 2Aa). All other stimuli led to significantly more attempts in both surface fish and cavefish (Fig 2Ab–2Ae). For the bead stimulus, as expected, surface fish were well engaged by showing more attempt numbers than water droplets under light conditions (both at the top and bottom of the recording arena; Fig 2Bb), but still responded to dark conditions (at the arena bottom; Fig 2Ab and 2Bb). This surface fish response in the light seems primarily driven by visual stimulus. Surface fish responses to the beads stimulus in the dark may be based on tactile or lateral line sensors since surface fish attempted to bite beads only close to or when touching beads (1–2 cm), which is the sensing range of tactile and lateral line sensors. Chemical sensing is not likely involved in the beads stimulus because beads did not emit food-like chemicals. Most surface fish mouthed beads, suggesting chemical stimulus—typically detected by extra mouth taste buds [23, 24]—is not necessary involved in capturing 'food'-like objects. Cavefish were less attracted to beads compared with surface fish (effect size, r = 0.66 compared with surface fish's r = 0.82; Fig 2Ab), but showed more attempts compared with water droplets (Fig 2Ab). Some cavefish showed a number of attempts at the top tank area in the dark (Fig 2Bb). Cavefish attempts in the top tank area could be based on similar reasons as latency: using auditory input and being bold in the dark. Cavefish did not show many attempts for beads in the bottom tank area under light or dark conditions (Fig 2Bb), suggesting cavefish may need additional stimuli, such as chemicals. In summary, cavefish may need further sensory inputs (integrating alternative sensory inputs) in addition to the object stimulus to maintain foraging behavior compared with surface fish.

Diet-extract chemical stimulus facilitated more attempts in both surface fish and cavefish, irrespective of light or dark conditions, than the beads stimulus (Fig 2Ac and 2Bc). These foraging attempts were mainly observed in the bottom tank area where food always sunk, suggesting fish may forage based on their previous experiences where the food always ended up.

For combined beads and food-extract stimulus, surface fish foraging patterns were similar to those observed in bead-only trials (see above; Fig 2Ad and 2Bd). However, cavefish

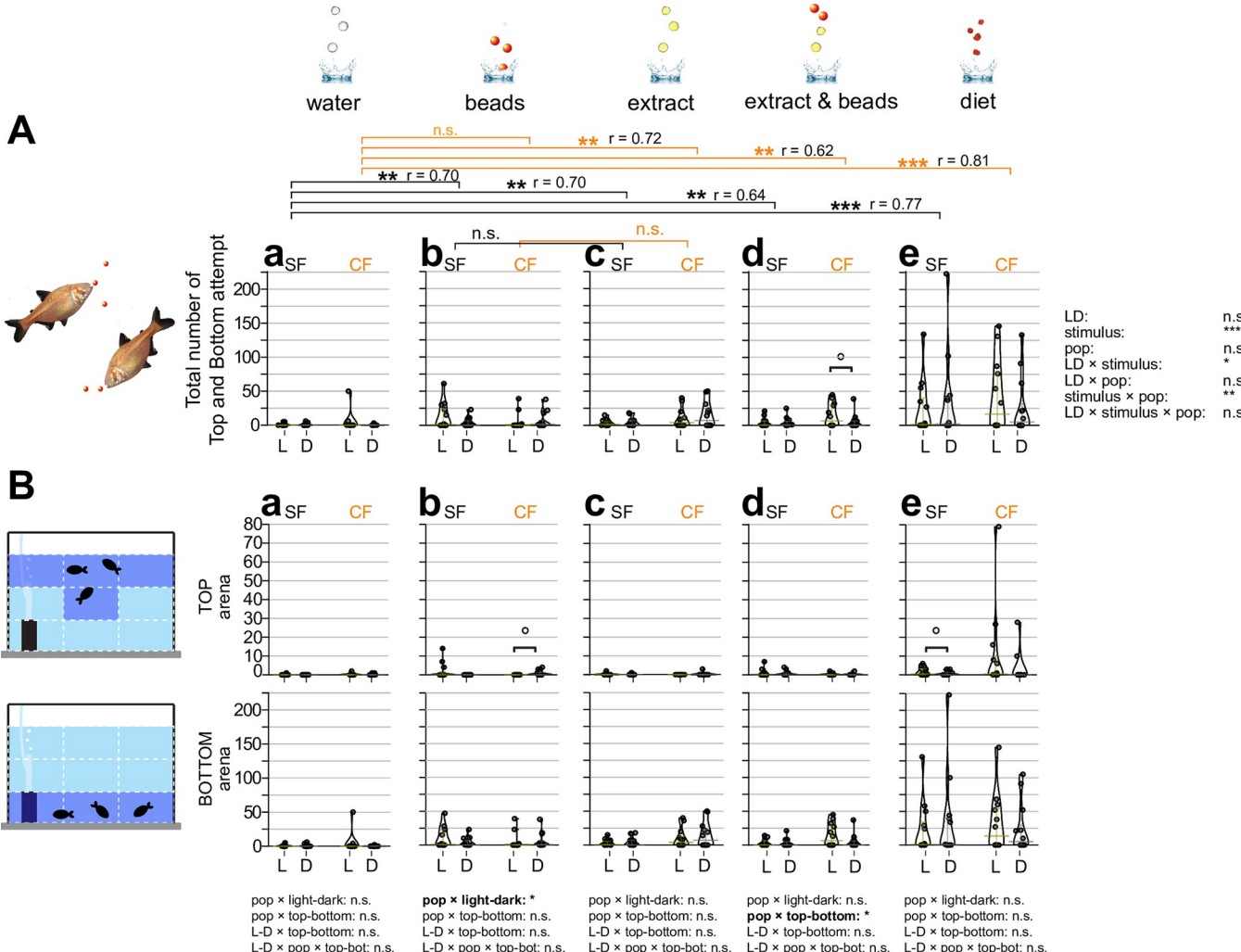

**Fig 2. Measured attempts responding to different sensory stimuli.** Overall attempt number in the 10-minute experiment defined as when fish obviously attempted a strike at the stimulus within the top or bottom areas. Three fish in a tank were given three droplets of RO purified water (water; **Aa** and **Ba**), three red plastic beads 4.7 mm in diameter (beads; **Ab** and **Bb**), three droplets of food extract (extract; **Ac** and **Bc**), three droplets of food extract followed by three red beads (extract and beads; **Ad** and **Bd**), and 3–4 granules (3–5 mm in diameter) of actual diet (diet; **Ae** and **Be**) (see Materials and Methods). In **Aa-Ae**, attempt (s) of surface fish (SF: left) and cavefish (CF: right) are plotted on the y-axis. Attempts under light condition (L: yellow bars and dots) and dark condition (D: gray bars and dots) are also shown. Statistical test result of the generalized linear model are shown on the far right (**A**). For each comparison, light and dark conditions were compared within the population per treatment as in Fig 1. Within each population, different stimuli were compared with the water stimulus and significances were calculated via Mann-Whitney tests adjusted by Holm's correction, shown as brackets at the top of the boxes. Comparisons between light and dark and between stimuli were significant. We also found significant differences when comparing light and dark responses and the stimuli and several interactions among the stimuli, populations, and light conditions. Details are available in S1 Table. (**B**) Fish locations were tracked as the top (top row) or bottom (bottom row) and measured attempts. The Y-axes and brackets represent the same as (**A**). All stars represent P-values after Holm's correction. Statistical test summaries using the generalized linear model including arena locations (top-bottom) are shown at the bottom of the boxes. Only interaction results are shown. Details of all statistics scores in this figure are in S1 Table. n.s.: not significant, ˚: P < 0.10, *: P < 0.05, **: P < 0.01, ***: P < 0.001.

increased their foraging attempts under light conditions, probably based on higher activity under ambient light [19]. This finding seemed to contradict those under the water and food-extract stimuli on latency (Fig 1Ba and 1Bc), where cavefish were more explorative under the dark. We consider that, because cavefish attempted at the 'bottom' instead of the top of the tank in the lighted condition (Fig 2Bd), they were not so explorative under the lighted condition. Also, compared with bead- and food extract-only trials, the combined stimulus with light may simultaneously facilitate foraging attempts where cavefish showed higher activities under

light. This notion was supported by food stimulus where cavefish also showed high attempts under light (Fig 2Ae and 2Be). For food stimulus, surface fish and cavefish were more active than other stimuli under both the light and dark conditions (Fig 2Ae and 2Be). The mechanism remains unclear. One possible explanation in the food stimulus trial is that the foraging sound of their cohorts (jaw clicking sound) evokes foraging behaviors in others [25]. Surface and cavefish may respond to such sounds strongly [26, 27]. This hypothesis requires further testing.

### Food discovery strategy (zigzag and circling motions)

Surface fish and cavefish showed specific movement patterns to locate stimulus (food), namely zigzag and circling motions (see Materials and Methods-Video Analysis). Both patterns were observed in surface fish and cavefish but used to varying degrees and in different contexts.

**Zigzag motion.** The zigzag motion was detected higher in cavefish (Figs 3 and 4), and particularly chemical stimulus (food extract, combined and diet stimulus) evoked longer zigzag dilation in cavefish (Fig 4Ac, 4Ad and 4Ae). This explorative behavior was also observed in surface fish more in the dark conditions (Figs 3Ac, 3Bd, 4Bc and 4Bd). Cavefish showed zigzag motion at the bottom in the light and explored the top in the dark (Figs 3Bd, 4Bc and 4Bd), suggesting that fish may be more free to express zigzag behavior in the areas that have a low risk of being found by predators in the wild. In summary, this zigzag motion is a shared response in surface and cavefish, primarily in the safer, darker areas.

**Circling motion.** The strong circling motion was observed with chemical stimulus as seen in the zigzag motion, but was more dominant in cavefish than surface fish (Figs 5 and 6). Cavefish exhibited high levels of circling motion under all stimuli compared with water droplet stimulus (Fig 5Aa–5Ae). These observations indicated that cavefish evolved circling motion to explore foods. Circling could be a better strategy than zigzagging given that circling yields fish come nearby the same food multiple times while only once while zigzagging.

### Conclusion

We examined foraging responses of surface and cavefish using water droplets (auditory stimulus), plastic beads (visual+auditory+lateral line+tactile), food extract (auditory+chemical), plastic beads & food extract, and actual food. We provided these stimuli in this order to avoid the chemical stimulus interrupt the beads stimulus. The full set of the experiment (N = 3 per population) was repeated two times by using different fish (N = 6 fish in total per population; 12 fish in total). To maximize foraging efficiency and minimize energy loss, visual/light conditions for surface fish favored beads and actual food (low latency; Fig 1) and surface fish captured these sources with a low number of attempts (Fig 2Ab, 2Ad, 2Ae, 2Bb, 2Bd and 2Be). Surface fish could also conserve energy by reducing total attempts toward non-visible objects (water droplets; Fig 2Aa and 2Ba). In contrast, in the dark, both surface and cavefish responded to auditory stimulus (water droplets; Fig 1Aa and 1Ba) to investigate without performing extra attempts (fewer attempts in water droplets; Fig 2Aa and 2Ba), which may be an efficient strategy to investigate objects if it is food. However, surface fish were less efficient with plastic beads by showing much higher attempts toward this inedible object (Fig 2Ab and 2Bb) than cavefish, suggesting visual stimulus is highly favored in foraging. In contrast, chemical stimulus evoked a higher number of attempts in cavefish than surface fish, indicating higher sensory emphasis on chemical sensing (olfaction and taste buds) for foraging in cavefish. This sensory priority in olfaction in cavefish is supported by the previous report indicating that cavefish responded to $10^5$ times lower concentrations of amino acid stimulus ($10^{-5}$ M vs $10^{-10}$ M of alanine in surface fish vs cavefish, respectively [14]. However, neither cavefish

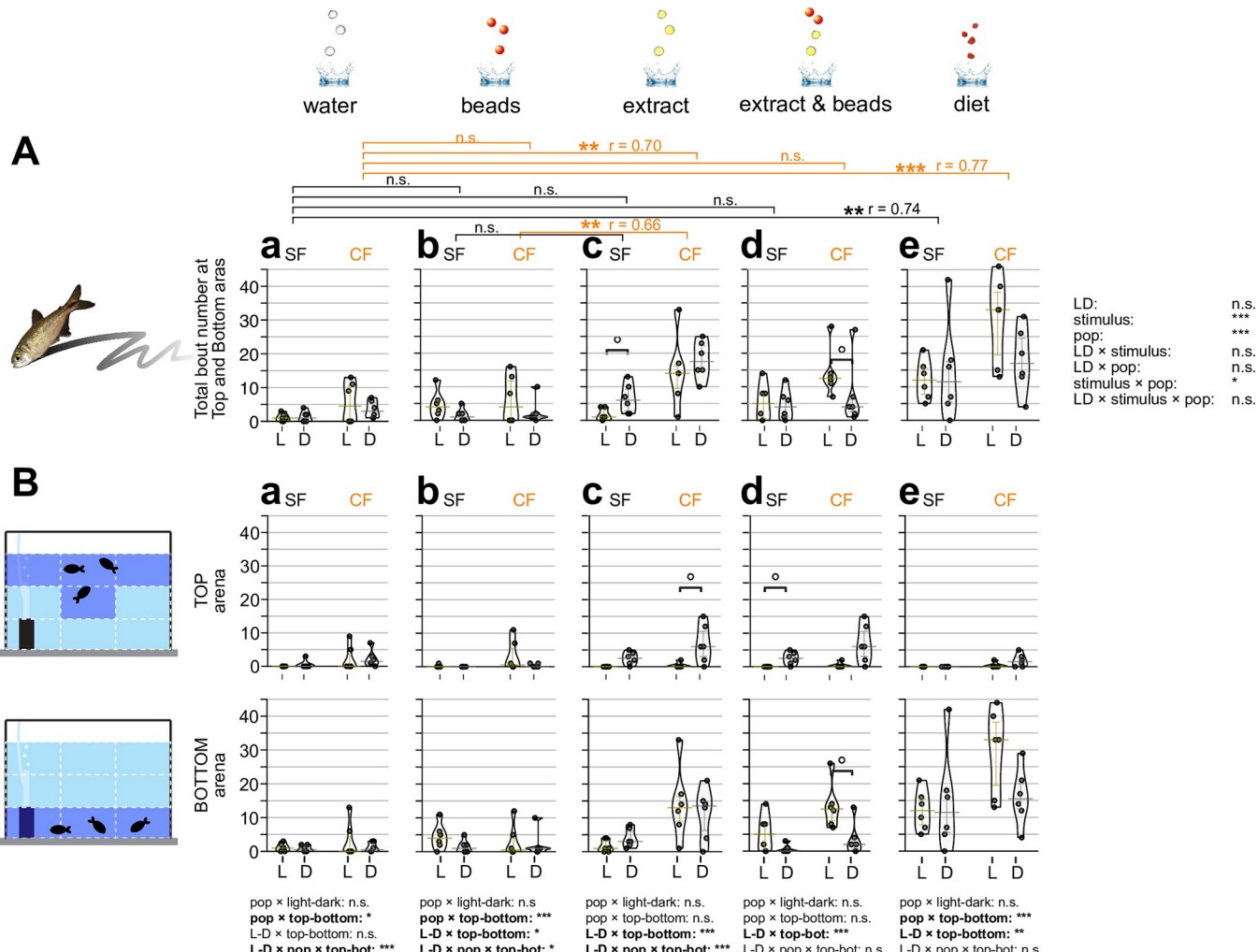

**Fig 3. Bout number of zigzag searching behavior in response to different sensory stimuli.** (**A**) Overall bout (i.e., event) counts for searching behavior using zigzag(s) in the 10-minute experiment. Zigzag searching behavior was defined as fish searching by zigzag motion (back and forth) frequently at the water surface or tank bottom with sensory stimuli (see Materials and Methods). The zigzag bout numbers of surface fish (SF: left) and cavefish (CF: right) are plotted on the y-axis. Zigzag behavior under light condition (L: yellow bars and dots) and dark condition (D: gray bars and dots) are also shown. Statistical test result of the generalized linear model is shown on the far right. For each comparison, light and dark conditions were compared within the population per treatment. Within each population, different stimuli were compared with water stimulus and significances were calculated via Mann-Whitney tests adjusted by Holm's correction (See S1 Table). (**B**) Fish locations were tracked as the top (top row) or bottom (bottom row) and measured zigzag behavior. The y-axes and brackets represent the same as (**A**). All stars represent P-values after Holm's correction. Statistical test summaries using the generalized linear model including arena locations (top-bottom) are shown at the bottom of the boxes. Only interaction results are shown. Details of all statistics scores in this figure are in S1 Table. n.s.: not significant, ˚: P < 0.10, *: P < 0.05, **: P < 0.01, ***: P < 0.001.

nor surface fish appeared to use chemical stimulus to navigate themselves toward food sources, as cavefish (and surface fish in the dark) started searching for food at the water surface or at the bottom immediately after <u>entering into</u> food extract clouds in the middle of the water column (S1 Movie). This seems to suggest that chemical stimulus indicates food presence instead of that fish use the odor gradient. This feeding strategy seems to contradict the previous reports where the chemical gradient looked to navigate *Astyanax* fish [14, 28]. However, we suspect that, while the chemical gradient informs the approximate direction that the fish must swim to approach the source of food in a still-water pool [28], the precise location of any suspended food particle is difficult to identify based on chemical sensing because of the slow

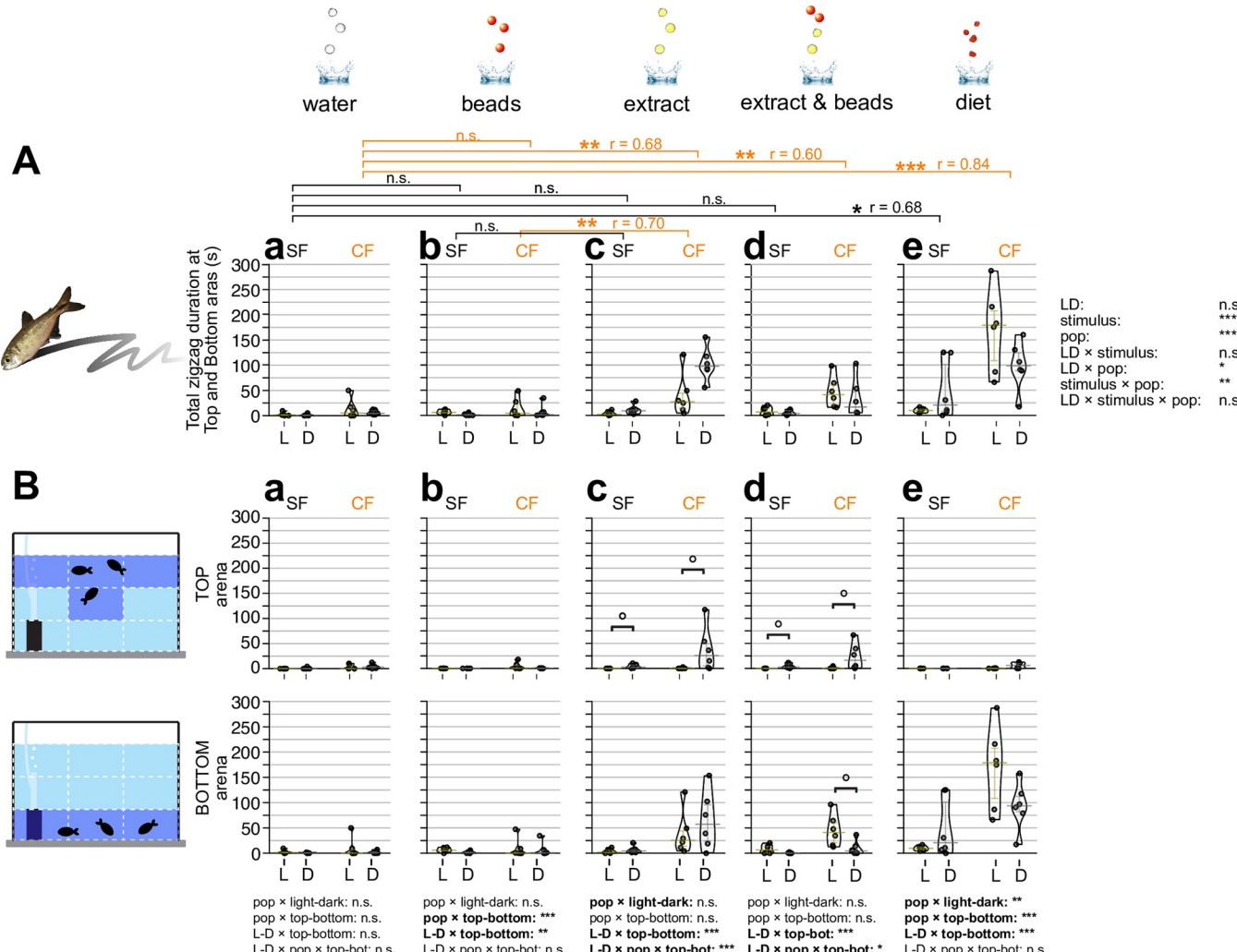

**Fig 4. Zigzag searching duration in response to different sensory stimuli.** (**A**) Overall searching duration (s) using zigzag(s) in the 10-minute experiment. Zigzag searching duration was measured when fish were searching with back-and-forth movements. The experimental setup was the same as Figs 1 and 3 (see Materials and Methods). The measured duration (s) of zigzag behavior of surface fish (SF: left) and cavefish (CF: right) are plotted on the y-axis in each panels (**Aa-Ae**). Zigzag behavior under light condition (L: yellow bars and dots) and dark condition (D: gray bars and dots) are also shown. Statistical test result of the generalized linear model is shown on the far right. For each panel, light and dark conditions were compared within the population per treatment. Within each population, different stimuli were compared with the water stimulus, and significances were calculated via Mann-Whitney tests adjusted by Holm's correction (See S1 Table). (**B**) Fish locations were tracked as the top (top row) or bottom (bottom row) and measured the zigzag behavior duration. The y-axes and brackets represent the same as (**A**). All stars represent P-values after Holm's correction. Statistical test summaries using the generalized linear model including arena locations (top-bottom) are shown at the bottom of the boxes. Only interaction results are shown. Details of all statistics scores in this figure are in S1 Table. n.s.: not significant, °: P < 0.10, *: P < 0.05, **: P < 0.01, ***: P < 0.001.

diffusion of molecules, which are advected by the fluid flow over a long time before they reach the fish's chemoreceptors. In contrast, the relatively fast diffusion of momentum through the viscous boundary layer around the fish enables particles near the boundary layer to be located quickly based on mechanical sensing [29]. Further study is needed to confirm this in a noisy environment.

Cavefish were more active by showing more attempts under light than dark when food scent was available (food extract and agar food), possibly due to higher activity under light [19] while foraging behavior was evoked by chemical stimulus (Fig 2Ad and 2Ae). We suspect this

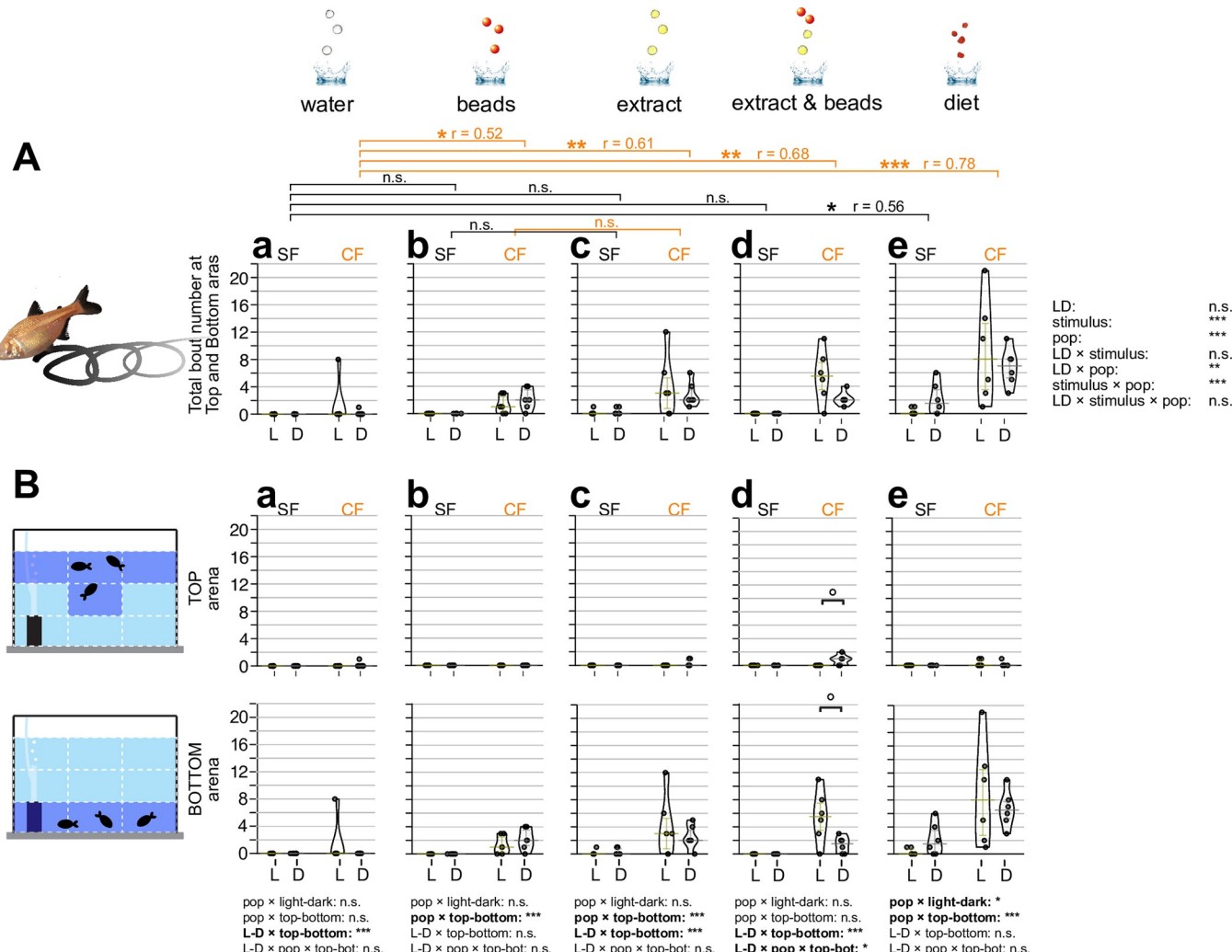

**Fig 5. Bout numbers of circling searching behavior in response to different sensory stimuli.** (**A**) Overall bout (i.e., event) numbers of circling motions fish during the 10-minute assay. Circling searching behavior is defined as fish repeating a circle pattern. The stimuli were given as in Fig 1 (see Materials and Methods too). The bout numbers of the circling motions of surface fish (SF: left) and cavefish (CF: right) were plotted on the y-axis during a 10-min observation in each panel of **Aa-Ae**. Circling behavior under light condition (L: yellow bars and dots) and dark condition (D: gray bars and dots) are also shown. Statistical test result of the generalized linear model is shown on the far right. For each comparison, the light and dark conditions were compared within the population per treatment. Within each population, different stimuli were compared with the water stimulus and significances were calculated via Mann-Whitney tests adjusted by Holm's correction (see S1 Table too). (**B**) Fish locations were tracked as the top (top row) or bottom (bottom row) and measured circling behavior. The y-axes and brackets represent the same as (**A**). All stars represent P-values after Holm's correction. Statistical test summaries using the generalized linear model including arena locations (top-bottom) are shown at the bottom of the boxes. Only interaction results are shown. Details of all statistics scores in this figure are in S1 Table. n.s.: not significant, °: P < 0.10, *: P < 0.05, ***: P < 0.001.

light-dependent response in cavefish is due to an evolutionary artifact of ambient light detection based on non-ocular opsins [19].

While both surface fish and cavefish showed similar levels of zigzag behavior (Figs 3 and 4), cavefish exhibited much more circling foraging than surface fish (Figs 5 and 6), suggesting circling may be an evolutionarily-enhanced strategy in cavefish, *i.e.* food could be less dispersed at the tank bottom compared with zigzagging, and also, cavefish have more chances to sense the same food multiple times compared with zigzagging, yielding only once in given time. As cavefish keep a similar level of zigzag behavior as surface fish, which probably makes fish explore larger areas than those of circling behavior, cavefish likely have a higher chance to

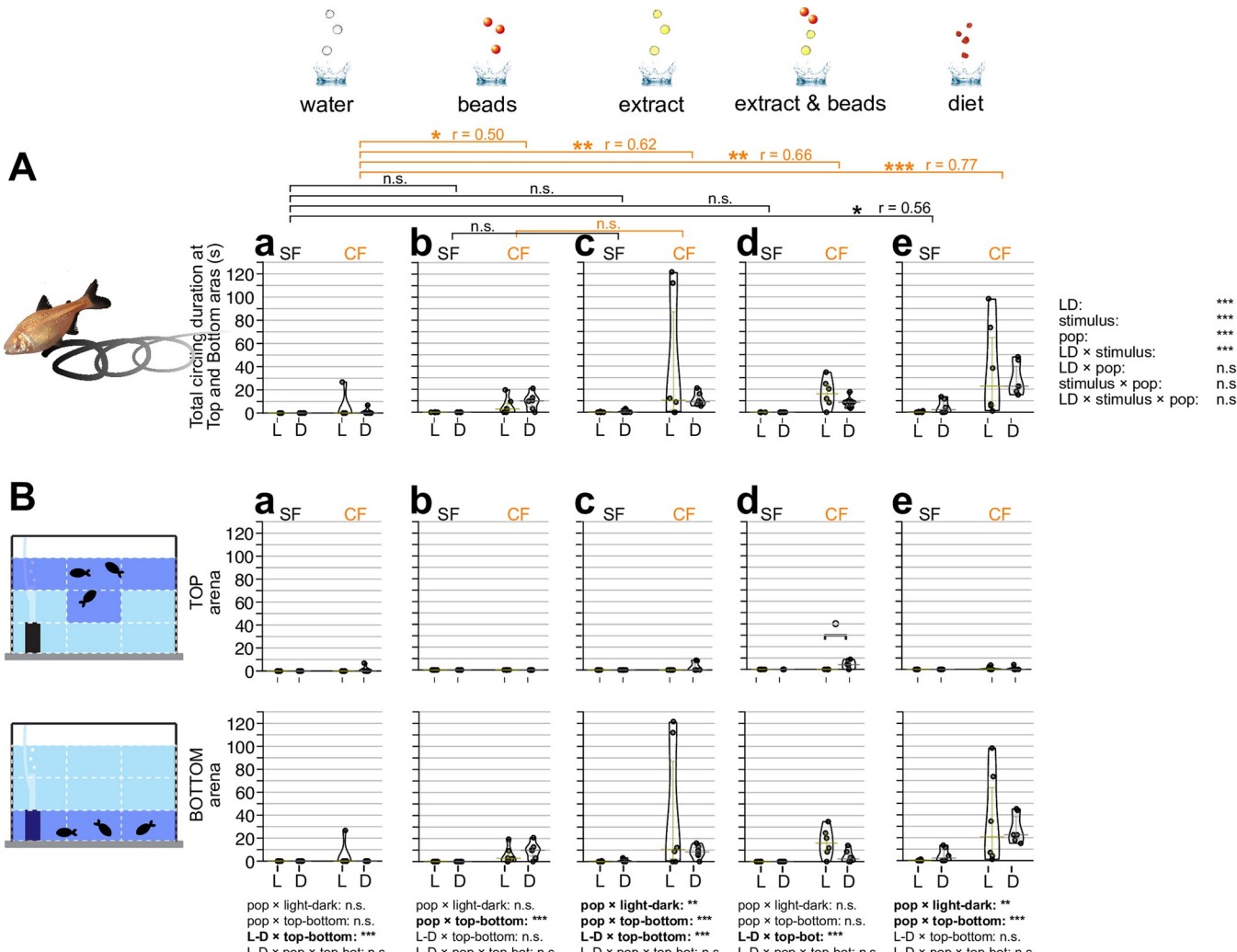

**Fig 6. Circling searching duration in response to different sensory stimuli.** (**A**) Overall duration of searching showing circling during the 10-minute observation. Circling searching duration is defined from when fish began searching in a repeated circle pattern to when fish stopped the behavior. Stimuli were given as in Figs 1 and 5 (see Materials and Methods). Duration of circling behavior of surface fish (SF: left) and cavefish (CF: right) were plotted on the y-axis within a 10 min observation in each panel of **Aa**-**Ae**. Circling behavior under light condition (L: yellow bars and dots) and dark condition (D: gray bars and dots) are also shown. Statistical test results of the generalized linear model are shown on the far right. For each panel, light and dark conditions were compared within the population per treatment. Within each population, different stimuli were compared with the water stimulus and significances were calculated via Mann-Whitney tests adjusted by Holm's correction, shown as brackets at the top of the boxes (see also S1 Table). (**B**) Fish locations were tracked as the top (top row) or bottom (bottom row) and measured circling behavior time. The y-axes and brackets represent the same as (**A**). All stars represent P-values after Holm's correction. Statistical test summaries using the generalized linear model including arena locations (top-bottom) are shown at the bottom of the boxes. Only interaction results are shown. Details of all statistics scores in this figure are in S1 Table. n.s.: not significant, °: P < 0.10, *: P < 0.05, **: P < 0.01, ***: P < 0.001.

reach food than surface fish by enhancing circling foraging. As for energy consumption of the muscle usages, we predict that zigzagging and circling are at similar levels, so there is no energetic advantage between these two foraging approaches. These ideas need further investigation to measure differences in foraging efficiency between zigzagging and circling.

## Supporting information

**S1 Table. Detailed statistical scores and summary of each figure.**
(XLSX)

**S1 Movie. An example movie exhibits the dispersal of methylene blue dye in the recording arena.**
(MOV)

## Acknowledgments

We are grateful to V Crystal, J Choi, L Lu, J Nguyen, C Balaan, K Lactaoen, M Worsham, H Hernandez, N Doeden, J Kato, M Ito, R Balmilero-Unciano, E Doy, A Martinez, D Mones, H Yoshizawa for fish care assistance. We also thank to M Iwashita for reviewing the final version of manuscript.

## Author Contributions

**Conceptualization:** Vânia Filipa Lima Fernandes, Daisuke Takagi, Masato Yoshizawa.

**Data curation:** Kyleigh Kuball, Vânia Filipa Lima Fernandes, Masato Yoshizawa.

**Formal analysis:** Kyleigh Kuball.

**Funding acquisition:** Daisuke Takagi, Masato Yoshizawa.

**Investigation:** Kyleigh Kuball, Vânia Filipa Lima Fernandes, Masato Yoshizawa.

**Methodology:** Kyleigh Kuball, Vânia Filipa Lima Fernandes, Masato Yoshizawa.

**Project administration:** Masato Yoshizawa.

**Resources:** Daisuke Takagi, Masato Yoshizawa.

**Software:** Masato Yoshizawa.

**Supervision:** Vânia Filipa Lima Fernandes, Daisuke Takagi, Masato Yoshizawa.

**Validation:** Vânia Filipa Lima Fernandes, Masato Yoshizawa.

**Visualization:** Masato Yoshizawa.

**Writing – original draft:** Masato Yoshizawa.

**Writing – review & editing:** Kyleigh Kuball, Vânia Filipa Lima Fernandes, Daisuke Takagi, Masato Yoshizawa.

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
