## [Decision Letter · Decision Letter 0]

28 Jul 2023

PONE-D-23-18287Blind cavefish evolved food-searching behavior without changing sensory modality compared with sighted conspecies in the darkPLOS ONE

Dear Dr. Yoshizawa,

Thank you for submitting your manuscript to PLOS ONE. After careful consideration, we feel that it has merit but does not fully meet PLOS ONE’s publication criteria as it currently stands. Therefore, we invite you to submit a revised version of the manuscript that addresses the points raised during the review process.

Please, address all the points raised by the two reviewers in a rebuttal letter, and upload a revised manuscript with track change. Given the nature of the points raised by reviewers, the revised manuscript will be sent back to them for re-review. In case you need more time to make revisions, please do not hesitate to ask for additional delay. In the case you do not wish to revise your manuscript, please inform us as well.  

We look forward to receiving your revised manuscript.

Kind regards,

Sylvie Rétaux

Academic Editor

PLOS ONE

Journal Requirements:

We gratefully acknowledge support from the National Institute of Health 

(P20GM125508) to MY, Hawaii Community Foundation (18CON-90818) to MY, and US Army 

Research Office (W911NF-17-1-0442) to DT. The funders had no role in study design, data 

collection and analysis, decision to publish, or preparation of the manuscript

However, funding information should not appear in the Acknowledgments section or other areas of your manuscript. We will only publish funding information present in the Funding Statement section of the online submission form. 

MY: National Institute of Health/ National Institute of General Medical Sciences (GM125508)

MY: Hawaii Community Foundation (18CON-90818)

DT: United States Army Research Office (W911NF-17-1-0442)

The funders had no role in study design, data collection and analysis, decision to publish, or preparation of the manuscript

4. We noted in your submission details that a portion of your manuscript may have been presented or published elsewhere. Please clarify whether this publication was peer-reviewed and formally published. If this work was previously peer-reviewed and published, in the cover letter please provide the reason that this work does not constitute dual publication and should be included in the current manuscript.

Reviewers' comments:

Reviewer's Responses to Questions

**Comments to the Author**

1. Is the manuscript technically sound, and do the data support the conclusions?

Reviewer #1: Partly

Reviewer #2: Partly

2. Has the statistical analysis been performed appropriately and rigorously? 

Reviewer #1: Yes

Reviewer #2: I Don't Know

3. Have the authors made all data underlying the findings in their manuscript fully available?

Reviewer #1: Yes

Reviewer #2: Yes

4. Is the manuscript presented in an intelligible fashion and written in standard English?

Reviewer #1: Yes

Reviewer #2: Yes

5. Review Comments to the Author

Reviewer #1: The present manuscript from Kuball et al., focuses on the evolution of food foraging strategies in the sighted-river and blind-cave morphotypes of the fish Astyanax mexicanus. The authors tested the implication of different sensory channels involved in food perception and localization i.e., vision, olfaction, audition and the lateral line, using a variety of original stimuli. The different stimuli were chosen in a way to stimulate specific sensory systems and used either individually or in combination, for example, red beads dropped in the water were supposed to trigger a visual response only, and the olfactory functions would also be recruited with the addition of food extracts. The authors have designed a complex set of sensory stimulation and performed an original description supported by glm and other conventional statistical analyses.

Overall, this study is original in the cavefish field and provides interesting insights into the evolution of the sensory strategies for food seeking.

However, the title and the abstract do not reflect some of the major differences reported in the introduction (e.g., line 83, 93) and elsewhere in the results. The authors found specific significant differences not only in the foraging strategy, but also in the immediate response to the different stimuli which deserve to be reflected in the title and in the abstract.

Additionally, the manuscript would benefit from more methodological details, a thorough description of the interesting results, and of a more elaborated discussion with environmental and evolutionary prospective on the foraging differences found by the authors. The manuscript would also benefit from a careful proofreading to allow a smoother reading. Comments and suggestions below.

Title and Abstract

The title and the abstract should better reflect the differences found in the results.

Line 34. “adherence level”. Please, explain what it is referring to, or use a more explicit term.

Introduction

Overall, the introduction would benefit from being more balanced: half of it describes the methodology, the results and some of the hypothesis. This could probably be moved to the Materials and Methods, and the Results or Discussion sections to lighten the Intro.

Since the study is about food seeking strategies it would be nice however, to see here comments and references to the differences in diet of those fish in regard to their environment, if any. Or some insights on the type of food available in the caves versus rivers to emphasize the importance of studying the evolution of foraging strategies.

Line 60. “promoted” is confusing in the middle of this sentence. Please, rephrase.

Line 82-85. This is a major difference in terms of foraging strategy between the two morphotypes. This should be relayed elsewhere, including in the abstract and title, and more discussed in the Results and Discussion. However, if the authors were making assumptions which are not validated in the results, this should be removed.

Line 93. “cavefish showed higher foraging…” Same comment as above. This is also an important difference which should be highlighted and more discussed elsewhere.

Materials and methods

This section should be more detailed according to the suggestions below.

Experimental populations.

This paragraph is not very clearly explained e.g., how long was the habituation period (3 days or 10min)? Could the authors clarify the difference between the 3 days of circadian training and the 10 min of acclimation? Could the authors provide precisions about the need to entrain the circadian rhythm of the fish?

Does the “2h ZT” correspond to the time when the fish are usually fed, and if so, would the authors expect an anticipation from the fish to receive food – could this have had influenced the responses to the experimental stimuli –?

This section mentions n=3 fish/tanks while the movie provided shows 6 fish in a tank. It would be interesting to discuss somewhere that the experiments were performed in groups of fish, and not in individual fish, which could have introduced a (social) bias in the responses to the different stimuli. It may be worth quickly considering previous studies mentioning that cavefish, in particular, tend to show a better motivation to forage/eat in social group.

Could the authors precise whether the five sessions of stimuli were randomized along the 50min or were always performed in the same sequential order? It would be necessary to mention/discuss this point in the Results and Discussion section.

If the water droplet assay is considered to be the positive control for triggering the audition solely, it would be worth discussing this bias in regard to all other stimuli which may also involve the auditory channel, and to take this into account in the interpretations.

It will be necessary to provide the number of technical replicates, or the number of tested groups (or the total fish number) for each experiment somewhere and to mention whether the fish were fed on their regular diet schedule, or starved before the trials.

Results and Discussion

Latency

Line 233-234. “… auditory stimulus was not sufficient…”

It would be interesting to discuss here and elsewhere, whether the water droplet test would also involve a response from the lateral line as mention in other studies (and not only the auditory system).

Line 263. “bold”. The concept of bold cavefish in the dark is interesting and tempting, but this term is a little bit surprising as risk-taking or anxiety (which could be considered as proxy for boldness) have not been tested here. It could be interesting to quickly elaborate how/why the dark condition would “disinhibit” the cavefish in the particular context of foraging?

Number of foraging attempts

Line 286. Please, quickly define the “effect size” and how is it calculated.

Line 324-326. “However, cavefish increased their foraging attempts under light…” it would be interesting to discuss this in regard to the previous statement suggesting a higher boldness in response to the water droplet under light condition.

Line 334. “Surface fish show more attempts in dark…” Do those increased attempts reflect an actual motivation to seek for food, or do they reflect the failure to capture the food extract?

Line 335-336. This paragraph seems to provide an interesting hypothesis which may deserve to be rephrased for a better understanding.

Food discovery strategy

Overall, more explanations and discussion of the results would be appreciated in this entire paragraph.

Line 250. “foraging sounds (agar food stimulus)”. This is a little confusing. Is sound the main stimulus supposed to be considered in the agar trial? Please, rephrase or clarify this sentence.

Conclusion

Line 407. Same comment as above about the water droplet stimulating the auditory system and potentially the lateral line as well.

Figures and legends

The figures are well made, easy to understand and the legends read well.

Reviewer #2: “Blind cavefish evolved food-searching behavior without changing sensory modality compared with sighted conspecifics in the dark” by Kuball et al investigates the response of surface and cave fish to different objects by providing different sensory cues to cavefish and surface fish. Through investigating latency to response, as well as 3 foraging-associated behaviors, the investigators determine how fish differentially respond to different potentially food-associated cues.

Major comments:

1. The study uses an n=3 for each condition/type of fish. First, this is a very low n, which may explain lack of significance between a number of conditions. Further, the 3 fish tested under each condition were tested within the same tank. As the authors point out that the fish may respond to other fish foraging, it is plausible that each fish’s behavior is not independent – If this is the case, it suggests that each condition is closer to n=1, making it hard to make firm conclusions from the presented results. Adding additional, independent assays, is needed to fully support these conclusions.

2. The authors assume the different stimuli are detected using different sets of sensory systems – however, they do not formally test this. For example, could droplets in the water be sensed by lateral line, not just auditory?

3. The argument that the fish do not follow gradients of chemical stimulus is not well supported by the data. Visualizing dye in the water does not necessarily mean that this is where the food particles are – it could be that there are some food particles in locations where the dye is too diffuse to see. Monitoring where food is in the water with an independent detector or using another method to ensure that there is a food gradient would be needed to make this argument.

Minor:

1. The argument that cavefish and surface fish can use similar sensory modalities to feed in the dark has been made before (for example, see Lloyd et al. 2018), and this conclusion should be discussed in the context of the literature. Further, as presented here, the data only suggest that cavefish and surface fish can respond to chemical cues in the dark, not that they use all of the same sensory modalities, nor that their food-seeking strategies did not need to evolve as they adapted to life in the dark – statements like the last line of the abstract should be removed as they are not supported by the data.

2. Throughout, there are statements that are not well supported by the data:

Line 84 – what data suggests that surface fish need multiple sensory inputs?

Line 95-96 – 2/3 surface fish also respond to water droplets.

Cavefish being bolder under dark conditions is highly speculative, and does not seem well supported by the data presented in figures 2-5.

Line 242 – Fig 1Bb top graph appears to contradict this statement.

Line 255 – More explanation needs to be given as to how this can be a sum of the other responses.

Line 268-269 – As cavefish have evolved differences in ability to detect food based on previous work cited by the authors, this seems to contradict what is here.

Line 285-286 – chemical stimulus could still be involved, it is just not required for this specific behavior with the beads.

Line 334 – These don’t appear to be different in the graph under light and dark conditions?

Line 348-349 – There do not appear to be differences in cavefish between light and dark from the graphs.

Statements regarding energy savings in the conclusions are highly speculative. Any statements about energy, as this was not tested, should be presented as speculative, and moved to the discussion, not the conclusions.

3. Line 170 – If attempts are defined as motions against a stimulus, how would this be defined for a food plume or a water droplet?

4. Figure 1B – more information is needed about what is being scored here.

5. All figures – why are 4 boxes designated top, while only 3 boxes are designated bottom?

6. Line 286-287 – What is being compared?

7. Line 319-320 – What is being compared?

8. Lines 337-338 are very speculative, I would recommend removing.

9. For the sections describing the last 4 figures, there are no references to individual figure panels, making this difficult to follow.

6. PLOS authors have the option to publish the peer review history of their article (what does this mean?). If published, this will include your full peer review and any attached files.

Reviewer #1: No

Reviewer #2: No

---

## [Author Response · Author response to Decision Letter 0]

21 Jan 2024

Reviewer #1: The present manuscript from Kuball et al., focuses on the evolution of food foraging strategies in the sighted-river and blind-cave morphotypes of the fish Astyanax mexicanus. The authors tested the implication of different sensory channels involved in food perception and localization i.e., vision, olfaction, audition and the lateral line, using a variety of original stimuli. The different stimuli were chosen in a way to stimulate specific sensory systems and used either individually or in combination, for example, red beads dropped in the water were supposed to trigger a visual response only, and the olfactory functions would also be recruited with the addition of food extracts. The authors have designed a complex set of sensory stimulation and performed an original description supported by glm and other conventional statistical analyses.

>Thank you for the positive summary on our study. Much appreciated. 

Overall, this study is original in the cavefish field and provides interesting insights into the evolution of the sensory strategies for food seeking. However, the title and the abstract do not reflect some of the major differences reported in the introduction (e.g., line 83, 93) and elsewhere in the results. The authors found specific significant differences not only in the foraging strategy, but also in the immediate response to the different stimuli which deserve to be reflected in the title and in the abstract.

Additionally, the manuscript would benefit from more methodological details, a thorough description of the interesting results, and of a more elaborated discussion with environmental and evolutionary prospective on the foraging differences found by the authors. The manuscript would also benefit from a careful proofreading to allow a smoother reading. Comments and suggestions below.

Title and Abstract

The title and the abstract should better reflect the differences found in the results.

>We worked on the title and abstract and the title now reads as “Blind cavefish evolved higher foraging responses through chemo- and mechanosensing”. Abstract contains new sentences L37 “Our results indicate cavefish …”

Line 34. “adherence level”. Please, explain what it is referring to, or use a more explicit term.

>Adherence level is “How much cavefish came to the stimulus (attempt) and engaged in searching (zigzag and circling)” we rewrote the sentence as “…affinity levels to the vibration stimuli were higher….” in the abstract.

Introduction

Overall, the introduction would benefit from being more balanced: half of it describes the methodology, the results and some of the hypothesis. This could probably be moved to the Materials and Methods, and the Results or Discussion sections to lighten the Intro.

>Thank you for pointing out. We did this arrangement in the introduction and make it clearer and concise the introduction

Since the study is about food seeking strategies it would be nice however, to see here comments and references to the differences in diet of those fish in regard to their environment, if any. Or some insights on the type of food available in the caves versus rivers to emphasize the importance of studying the evolution of foraging strategies.

>We made it clearer in possible ecological relevance in the introduction.

Line 60. “promoted” is confusing in the middle of this sentence. Please, rephrase.

>Thank you for pointing this out. It reads now as “… and cavefish enhanced this response by the increased cranial mechanosensory lateral line”.

Line 82-85. This is a major difference in terms of foraging strategy between the two morphotypes. This should be relayed elsewhere, including in the abstract and title, and more discussed in the Results and Discussion. However, if the authors were making assumptions which are not validated in the results, this should be removed.

>Thank you for pointing this too. Our new result indicated that some of surface fish responded to sole auditory stimulus too. We made it clear then.

Line 93. “cavefish showed higher foraging…” Same comment as above. This is also an important difference which should be highlighted and more discussed elsewhere.

>Thank you for pointing out too. We made it clearer in Result, discussion and abstract.

Materials and methods

This section should be more detailed according to the suggestions below.

Experimental populations.

This paragraph is not very clearly explained e.g., how long was the habituation period (3 days or 10min)? Could the authors clarify the difference between the 3 days of circadian training and the 10 min of acclimation? Could the authors provide precisions about the need to entrain the circadian rhythm of the fish?

>Sorry for our uncleanness. We rewrote the sentences and made it clearler as “Fish in the cleaned tank were (four days prior to recording) then placed on the recording stages in a light-controlled room where fish circadian rhythm was entrained by a 12:12 h light-dark cycle with 30–100 lux light. On recording days, the experiment commenced at ~2 hours of Zeitgeber time. We then set recording cameras (see below), set blackboards on the side of the arena to prevent extra visual stimulus, and waited for a 10-min acclimation time prior to recording.”

Does the “2h ZT” correspond to the time when the fish are usually fed, and if so, would the authors expect an anticipation from the fish to receive food – could this have had influenced the responses to the experimental stimuli –?

>The reviewer is correct that ZT2 is approximately the same time as every day’s feeding time (ZT3) to provide a higher chance of exhibiting foraging behaviors. However, we did not observe the change in fish behavior, such as searching for food before the recording. We think this recording schedule well standardized the experiment.

This section mentions n=3 fish/tanks while the movie provided shows 6 fish in a tank. It would be interesting to discuss somewhere that the experiments were performed in groups of fish, and not in individual fish, which could have introduced a (social) bias in the responses to the different stimuli. It may be worth quickly considering previous studies mentioning that cavefish, in particular, tend to show a better motivation to forage/eat in social group.

>sorry for the confusion. The video supplied as a supplemental data was old and all of experiments were performed 3 fish in a tank. In this revision we added another independent 3 fish from the original setup therefore n = 6 (two experimental replicates). This data will support our conclusion better.

Could the authors precise whether the five sessions of stimuli were randomized along the 50min or were always performed in the same sequential order? It would be necessary to mention/discuss this point in the Results and Discussion section.

>It was in the same order due to the effect of residual of scent (food extract), which was hard to remove from the tank without disturbing fish. We mentioned it in the Materials and Method section.

If the water droplet assay is considered to be the positive control for triggering the audition solely, it would be worth discussing this bias in regard to all other stimuli which may also involve the auditory channel, and to take this into account in the interpretations.

>thank you for this input. This is great point. We used the water droplets as the baseline of the experiment and mentioned it in our Result & Discussion section now.

It will be necessary to provide the number of technical replicates, or the number of tested groups (or the total fish number) for each experiment somewhere and to mention whether the fish were fed on their regular diet schedule, or starved before the trials.

>Thank you for pointing out this. Now these are mentioned in the “Experimental populations” section in the Materials and Methods.

Results and Discussion

Latency

Line 233-234. “… auditory stimulus was not sufficient…”

It would be interesting to discuss here and elsewhere, whether the water droplet test would also involve a response from the lateral line as mention in other studies (and not only the auditory system).

>Accordingly, we added a discussion about the involvement of the lateral line sensor in detection of the water droplet as “The water droplet stimulus could be detected by the inner ear and less likely by the mechanosensory lateral line system because, typically, the lateral line could sense ~1.5× of the body length distance[17], which is ~ 10 cm away from Astyanax fish. The inner ear can sense much further distance[18].”

Line 263. “bold”. The concept of bold cavefish in the dark is interesting and tempting, but this term is a little bit surprising as risk-taking or anxiety (which could be considered as proxy for boldness) have not been tested here. It could be interesting to quickly elaborate how/why the dark condition would “disinhibit” the cavefish in the particular context of foraging?

>The reviewer is correct that we over-stated here without testing the anxiety level. We changed the word as “explorative” instead of “bold”. 

Number of foraging attempts

Line 286. Please, quickly define the “effect size” and how is it calculated.

>In statistics, “effect size” is the magnitude of the difference between groups. In other words, “effect size” is what extent of the variance in the given measurements (latency duration) the population categories explain. Also explained it in the Materials and Methods.

Line 324-326. “However, cavefish increased their foraging attempts under light…” it would be interesting to discuss this in regard to the previous statement suggesting a higher boldness in response to the water droplet under light condition.

>This is quite interesting point and we thank to the reviewer. The multiple sensory inputs from beads and food-extract may further enhance the foraging attempts. We therefore added a sentence to clarify this point as “This finding seemed to contradict those under the water and food-extract stimuli on latency (Fig. 1Ba and Bc), where cavefish were more explorative under the dark. We consider that, because cavefish attempted at the ‘bottom’ instead of the top of the tank in the lighted condition (Fig. 2Bd), they were not so explorative under the lighted condition.”

Line 334. “Surface fish show more attempts in dark…” Do those increased attempts reflect an actual motivation to seek for food, or do they reflect the failure to capture the food extract?

>This is great point and we included this possibility as well. 

Line 335-336. This paragraph seems to provide an interesting hypothesis which may deserve to be rephrased for a better understanding.

>New data indicated that this point is not valid anymore. We rewrote the section according to the updated data.

Food discovery strategy

Overall, more explanations and discussion of the results would be appreciated in this entire paragraph.

Line 250. “foraging sounds (agar food stimulus)”. This is a little confusing. Is sound the main stimulus supposed to be considered in the agar trial? Please, rephrase or clarify this sentence.

>This statement was removed due to the change of the result in zigzag behavior. Essentially the major point for zigzag motion did not change.

Conclusion

Line 407. Same comment as above about the water droplet stimulating the auditory system and potentially the lateral line as well.

>Accordingly, we have enriched the conclusion statement by following reviewer’s comment.

Figures and legends

The figures are well made, easy to understand and the legends read well.

Reviewer #2: “Blind cavefish evolved food-searching behavior without changing sensory modality compared with sighted conspecifics in the dark” by Kuball et al investigates the response of surface and cave fish to different objects by providing different sensory cues to cavefish and surface fish. Through investigating latency to response, as well as 3 foraging-associated behaviors, the investigators determine how fish differentially respond to different potentially food-associated cues.

Major comments:

1. The study uses an n=3 for each condition/type of fish. First, this is a very low n, which may explain lack of significance between a number of conditions. Further, the 3 fish tested under each condition were tested within the same tank. As the authors point out that the fish may respond to other fish foraging, it is plausible that each fish’s behavior is not independent – If this is the case, it suggests that each condition is closer to n=1, making it hard to make firm conclusions from the presented results. Adding additional, independent assays, is needed to fully support these conclusions.

>In response to their point, we have added another 3-fish set of experiment, gaining statistical power. As pointed, each fish may affect their behavior responses. We think that this repeated data averaged out the effect of cohort fish (two other fish).

2. The authors assume the different stimuli are detected using different sets of sensory systems – however, they do not formally test this. For example, could droplets in the water be sensed by lateral line, not just auditory?

>The reviewer is correct that we did not test each sensor. However, testing each sensor is beyond the scope of this manuscript. We describe and discuss the ‘possible’ sensory systems involved in sensing each stimulus in the Introduction and Result sections.

3. The argument that the fish do not follow gradients of chemical stimulus is not well supported by the data. Visualizing dye in the water does not necessarily mean that this is where the food particles are – it could be that there are some food particles in locations where the dye is too diffuse to see. Monitoring where food is in the water with an independent detector or using another method to ensure that there is a food gradient would be needed to make this argument.

>First of all, the food extract is filtered by the 0.45 um syringe filter, which filters many bacteria and all particles that sink quickly in water. Through the view of the fluid dynamics, such small particles, including food scents (amino acids, small peptides, fatty acids, and sugars), behave as other solubilized compounds such as methylene blue. We agree with the reviewer that the dye would not be visible in locations where the food particles are too diffuse to see. However, where the food is more concentrated, the dye's color intensity is indicative of the concentration field. During the short time interval between the dye injection and fish's response, our visualization revealed slow downward advection and diffusion of food particles. The concentration decreased towards the bottom of the tank, which supports our claim that the bottom-aiming fish did not follow gradients in chemical stimulus.

Minor:

1. The argument that cavefish and surface fish can use similar sensory modalities to feed in the dark has been made before (for example, see Lloyd et al. 2018), and this conclusion should be discussed in the context of the literature. Further, as presented here, the data only suggest that cavefish and surface fish can respond to chemical cues in the dark, not that they use all of the same sensory modalities, nor that their food-seeking strategies did not need to evolve as they adapted to life in the dark – statements like the last line of the abstract should be removed as they are not supported by the data.

>As we added data and reviewer#1’s comment, we updated the conclusion as ‘we propose ancestors of cavefish similar to the modern surface fish evolved extended food-seeking behaviors, including circling motion, to adapt to the dark’ in the abstract, and similar statements in Result & Discussion section. Also, we added Lloyd et al. 2018 reference.

2. Throughout, there are statements that are not well supported by the data:

Line 84 – what data suggests that surface fish need multiple sensory inputs?

>According to the updated data, it is now read as “some surface fish and cavefish responded to sole auditory stimulus (water droplet) in either light or dark conditions, but their response became more robust with visual (beads: for surface fish) or chemical (food scent: for cavefish) stimulus, suggesting both fish rely on multiple sensory inputs for the initial response (latency).”

Line 95-96 – 2/3 surface fish also respond to water droplets.

>Now it reads as “surface fish tended to require multiple sensory stimuli to engage to forage”

Cavefish being bolder under dark conditions is highly speculative, and does not seem well supported by the data presented in figures 2-5.

>We agree that ‘bolder’ was an overstatement. Now it reads as “dark conditions seemed to make cavefish more explorative to come to the water surface.”

Line 242 – Fig 1Bb top graph appears to contradict this statement.

>The updated data supports the original statement: “Cavefish responded to beads similarly to surface fish in the dark irrespective of light or dark conditions (Fig 1Bb).”

Line 255 – More explanation needs to be given as to how this can be a sum of the other responses.

>The procedure how we applied the stimuli (beads and food-extract) were explained in the Materials and Methods. We agreed that we needed more explanation of our statement and now it reads as “The combined bead and food-extract stimulus (Fig. 1Ad) invoked the intermediate response of beads-only and food extract-only stimulus in cavefish, where cavefish responded to the stimulus by aiming to the bottom under the light condition and aimed at either the top or bottom under the dark condition (Fig 1Bd).”

Line 268-269 – As cavefish have evolved differences in ability to detect food based on previous work cited by the authors, this seems to contradict what is here.

>We do not agree with the reviewer’s statement although we understand her/his point of the prior knowledge in the sensory system. Our conclusion is restricted to the “initial foraging response” and based on what we found. We consider that we do not need to change our statement “…, suggesting cavefish did not evolve particular sensory responses during initial foraging attempts in the dark.”

Line 285-286 – chemical stimulus could still be involved, it is just not required for this specific behavior with the beads.

>In the context of this statement, we concluded that chemical stimulus is not likely to be involved in mouthing beads because the beads were washed overnight with fresh fish water, and cavefish (having better chemical sensing ability than surface fish) did not mouth the beads. We decided to keep our conclusion as it was: “Chemical sensing is not likely involved here because beads did not emit food-like chemicals. Most surface fish mouthed beads, suggesting chemical stimulus—typically detected by extra mouth taste buds [22,23]—is not necessary involved in capturing ‘food’-like objects.”

Line 334 – These don’t appear to be different in the graph under light and dark conditions?

>Thank you for pointing out. The reviewer is correct that surface fish did not show higher attempts in the dark than light condition with food stimulus. We then restated as “For food stimulus, surface fish and cavefish were more active than other stimuli under both the light and dark conditions (Fig 2Ae and 2Be).”

Line 348-349 – There do not appear to be differences in cavefish between light and dark from the graphs.

Statements regarding energy savings in the conclusions are highly speculative. Any statements about energy, as this was not tested, should be presented as speculative, and moved to the discussion, not the conclusions.

>L-D difference disappeared after adding new data and we updated the statement – surface fish and cavefish tended to express zigzag explorative motion in the safer condition (i.e., dark for cavefish or bottom for both surface fish and cavefish). We moved the speculation-statements from our conclusions. 

3. Line 170 – If attempts are defined as motions against a stimulus, how would this be defined for a food plume or a water droplet?

>Food plume and water droplet were colored with methylene blue to track their locations. This is described in Materials and Methods.

4. Figure 1B – more information is needed about what is being scored here.

>Scoring procedure was the same as Fig 1A. To clarify, we added a sentence in the fig legend as “Fish locations were tracked as the top (top row) or bottom (bottom row) and measured latencies as the same as Fig 1A”.

5. All figures – why are 4 boxes designated top, while only 3 boxes are designated bottom?

>We are afraid that this inconsistency came from the conversion error. We checked it carefully in this revision. 

6. Line 286-287 – What is being compared?

>Thank you for pointing out. Now it reads as “Cavefish were less attracted to beads compared with surface fish (effect size, r = 0.66 compared with surface fish’s r = 0.82; Fig 2Ab)”

7. Line 319-320 – What is being compared?

>Now it reads as “Diet-extract chemical stimulus facilitated more attempts in both surface fish and cavefish irrespective of light or dark conditions than the beads stimulus”

8. Lines 337-338 are very speculative, I would recommend removing.

>We removed this sentence accordingly.

9. For the sections describing the last 4 figures, there are no references to individual figure panels, making this difficult to follow.

>The explanation for Fig 3-6 are the repeat of Fig 1 and 2 but measuring the different behaviors (zigzaging and circling). We believe the important messages are all included. Also we added more sentences for clarity.

---

## [Decision Letter · Decision Letter 1]

6 Mar 2024

Blind cavefish evolved higher foraging responses to chemo- and mechanostimuli

PONE-D-23-18287R1

Dear Dr. Yoshizawa,

We’re pleased to inform you that your manuscript has been judged scientifically suitable for publication and will be formally accepted for publication once it meets all outstanding technical requirements. The revised version has been reviewed by one of the previous reviewers and myself, and we both agree that the ms has been much improved.

Kind regards,

Sylvie Rétaux

Academic Editor

PLOS ONE

Additional Editor Comments (optional):

Reviewers' comments:

Reviewer's Responses to Questions

**Comments to the Author**

1. If the authors have adequately addressed your comments raised in a previous round of review and you feel that this manuscript is now acceptable for publication, you may indicate that here to bypass the “Comments to the Author” section, enter your conflict of interest statement in the “Confidential to Editor” section, and submit your "Accept" recommendation.

Reviewer #1: All comments have been addressed

2. Is the manuscript technically sound, and do the data support the conclusions?

Reviewer #1: Yes

3. Has the statistical analysis been performed appropriately and rigorously? 

Reviewer #1: Yes

4. Have the authors made all data underlying the findings in their manuscript fully available?

Reviewer #1: Yes

5. Is the manuscript presented in an intelligible fashion and written in standard English?

Reviewer #1: Yes

6. Review Comments to the Author

Reviewer #1: In this revised version of the manuscript, the authors have adequately addressed all previous comments. The authors have added experiments to increase the sample number, hence improving the statistical power. They have changed some interpretations and conclusions accordingly. The authors have made significant efforts to comply with the comments and are presenting an original study.

7. PLOS authors have the option to publish the peer review history of their article (what does this mean?). If published, this will include your full peer review and any attached files.

Reviewer #1: No

---

## [Editor Report · Acceptance letter]

12 Mar 2024

PONE-D-23-18287R1 

PLOS ONE

Dear Dr. Yoshizawa, 

I'm pleased to inform you that your manuscript has been deemed suitable for publication in PLOS ONE. Congratulations! Your manuscript is now being handed over to our production team.

Kind regards, 

on behalf of

Dr. Sylvie Rétaux 

Academic Editor

PLOS ONE